



# 1 Lagrangian Tracking of Moisture Sources for the Record-Breaking

# 2 Rainfall of Storm Ianos

Patricia Coll-Hidalgo[1], Raquel Nieto[1,2], Alexandre Ramos[3], Patrick Ludwig[3], Luis Gimeno[1,2]
[1]Centro de Investigación Mariña, Universidade de Vigo, Environmental Physics Laboratory (EPhysLab), Campus As Lagoas
s/n, 32004 Ourense, Spain.
[2]Galicia Supercomputing Center (CESGA), Santiago de Compostela, Spain
[3]Institute of Meteorology and Climate Research (IMKTRO), Karlsruhe Institute of Technology (KIT), Karlsruhe, Germany
*Correspondence to*: Patricia Coll-Hidalgo (patricia.coll@uvigo.es)
**Abstract.**
This study utilizes a Lagrangian moisture tracking approach, supported by high-resolution weather simulations, to identify
and quantify the sources of moisture contributing to precipitation associated with storm Ianos in the Mediterranean in
September 2020. The findings reveal that the Ionian Basin and the southwestern Balkan Peninsula were the primary moisture
contributors, closely aligned with the cyclone's trajectory. Secondary sources included regions in North Africa, such as
Libya and Tunisia, the Tyrrhenian Basin, southern Italy, the Aegean, Marmara, and the Black Seas.
Moisture transport occurred along three dominant pathways. The first originated from the Black Sea, passing over the
Marmara Sea and entering the region between Greece and the Dodecanese Islands. The second pathway traced particles from
the Tyrrhenian Basin, across the Algerian Basin and Libya, before reaching the storm's core. The third route extended
eastward from Northwest Africa, crossing the Gulf of Gabes. Among these, the Marmara-Black Sea region emerged as the
most significant remote source, contributing moisture from the surface to approximately 850 hPa.
As Storm Ianos intensified, moisture flux from remote sources increased, with the final 24 hours before landfall marking the
most significant period of moisture uptake. During this critical phase, in-situ evaporative processes over the Greek coastline
and the Ionian Sea became dominant, with the last 36 hours contributing the majority of precipitation-related moisture.

## 1 Introduction

The Mediterranean basin (Fig. 1) is a hotspot for extratropical cyclones, encompassing a wide spectrum in their cyclogenesis
processes, deepening rates, and geographical origins (Flaounas et al., 2022). These Mediterranean cyclones play a critical
role in the region's hydrological cycle, significantly impacting annual precipitation and contributing to extreme rainfall
events (Pfahl and Wernli, 2012; Michaelides et al., 2018). In addition to their direct influence on precipitation,





Mediterranean cyclones also affect the water budget of the region by intensifying evaporative processes and facilitating
atmospheric moisture fluxes from external regions, such as the Eastern North Atlantic Ocean (Flaounas et al., 2016, 2018).
The sources of moisture for cyclones precipitation have been widely studied, with key contributions from the Mediterranean
Sea itself, as well as from remote sources such as the tropical and extratropical Atlantic Oceans and tropical Africa
(Winschall et al., 2014; Chazette et al., 2016; Lee et al., 2017; Raveh-Rubin and Wernli, 2016; Duffourg et al., 2018).
Notably, Raveh-Rubin and Wernli (2016) found that intense Mediterranean cyclones are typically fueled by a combination
of local Mediterranean moisture and secondary moisture from more distant regions. Flaounas et al. (2019) further classified
Mediterranean cyclones based on their rainfall intensity, revealing that cyclones producing high rainfall are typically
associated with greater moisture contributions from the eastern Atlantic and the Mediterranean basin, particularly the
western Mediterranean Sea.
The availability of satellite imagery has enabled the detection of cyclones in the Mediterranean Sea with unusual features,
such as a cloud-free region at the centre (eye) and a high axial symmetry (Ernst and Matson, 1983; Rasmussen and Zick,
1987). These infrequent events, referred to as Mediterranean hurricanes (medicanes) or tropical-like cyclones (TLCs),
display additional characteristics that distinguish them from both extratropical and tropical low-pressure systems. The
genesis of medicanes occurs in a baroclinic environment and eventually evolves into warm core vortices during their mature
stage, which represents a brief period within their typically short lifespan (Miglietta and Rotunno, 2019; Gutiérrez-
Fernández et al., 2024). The occurrence of tropical transitions and intensification in these systems exhibits significant case-
to-case variability, although it is primarily driven by processes of diabatic forcing and baroclinic instability (Miglietta and
Rotunno, 2019; Dafis et al., 2020; Flaounas et al., 2021; Flaounas et al., 2022).
They can produce unusually high rainfall throughout their lifecycle (Flaounas et al., 2018). Zhang et al. (2020) observe that
rainfall totals increase from the centre to approximately 0.8 degrees and then decrease. Dafis et al. (2020) emphasize that
TLC intense convective activity isn't sustained close to the cyclone centre for very long; only a subset of medicanes has
long-lasting, intense convection near their core. For example, in the rare case of one of the strongest medicanes, Storm Ianos
(September 2020), Lavaguardos et al. (2021) identified deep convection in the proximity of the medicane core during the
mature phase. The authors also linked the record-breaking rainfall associated with this medicane to unusually high levels of
effective precipitable water over the Mediterranean, underscoring the importance of moisture availability in determining the
intensity of rainfall from these systems.
Storm Ianos (September 2020) stands out due to several distinct characteristics. Its central pressure reached 984.3 hPa,
accompanied by maximum wind speeds of 54 m s$^{-1}$ recorded in Palliki, Cephalonia, Greece. In terms of precipitation, daily
accumulated rainfall exceeded 600 mm in western Greece, notably in the northernmost region of Cephalonia (Antipata), with
central Greece also experiencing significant rainfall totals of over 300 mm (Lagouvardos et al., 2021). The storm also caused
significant impacts, including over 1,400 landslides (Zekkos et al., 2020) and intense lightning activity, with peak flash rates
surpassing typical medicane values (D'Adderio et al., 2022). The medicane caused significant storm surges, with peaks of
0.19 m at Zakynthos, 0.27 m at Katakolo, and 0.16 m at Kyparissia, all located in western Greece. Meanwhile, the Pylos




buoy, situated off the southwestern coast of the Peloponnese, recorded a maximum significant wave height of 4.7 m at 03:00 UTC on 18 September, although it likely missed the peak of the storm (Ferrarin et al., 2023). Flash flooding caused the tragic deaths of four people, with the Fire Service responding to 630 calls, performing 450 rescues and 120 flood water pumping operations. Significant damage occurred in central Greece, particularly in Farsala, Mouzaki, and Karditsa, where agricultural and urban areas were severely affected. The village of Assos, in Cephalonia, was buried under debris. Several bridges collapsed, and transport networks were severely disrupted. Economic and insured losses are estimated at USD 100 million (Zekkos et al., 2020).

Despite the documented extreme weather associated with Storm Ianos, the precise sources of moisture that fueled its unprecedented precipitation remain unclear. To address this gap, the present study aims to identify the moisture sources responsible for the rainfall associated with Ianos using a Lagrangian tracking approach and high-resolution data.

## 2 Data and Methods

For this study, dynamic downscaling of ERA5 reanalysis data from the European Centre for Medium-Range Weather Forecasts (ECMWF; Hersbach et al., 2020) was performed using the Weather Research and Forecasting (WRF; Skamarock et al., 2008) mesoscale model. The high-resolution output from WRF was subsequently used as input for the Lagrangian particle dispersion model, facilitating detailed analysis of moisture transport and source identification. The subsequent processing of this data is outlined in the following sub-sections.

### 2.1 Models Design and Setup

(a) WRF model

The WRF model (v4.2) was used to simulate a 25-day period, including 7 days as model spin-up (Jerez et al., 2020; Liu et al., 2023), starting on 1st September 2020 at 00:00 UTC until 25th September 2020 at 00:00 UTC. Boundary conditions were updated every 6 hours using ERA5 reanalysis data. The simulation featured two nested domains: a coarser grid with a horizontal resolution of 18 km and a finer grid with a horizontal resolution of 6 km (Fig. 1a). Both domains included 40 vertical levels, with the model top set to 50 hPa.

For physical parameterizations, the WRF model used the WSM6 scheme (Hong and Lim, 2006) for microphysics, the Rapid Radiative Transfer Model (RRTM) for longwave radiation (Mlawer et al., 1997), and the Dudhia scheme for shortwave radiation (Dudhia, 1989). The Noah Land Surface Model (Chen and Dudhia, 2001) was employed for surface-atmosphere interactions, while the YSU planetary boundary layer (PBL) scheme (Hong et al., 2006) and the Kain-Fritsch scheme (Kain, 2004) were used for boundary layer and convection processes, respectively.

To ensure consistency with large-scale atmospheric patterns, we applied gridded nudging (every 6 h) to both domain grids. Nudging was implemented at multiple vertical levels, and to assess the impact of vertical nudging, several experiments were





conducted. These included: nudging applied throughout the entire vertical column (WRF_full_ndg), a configuration with no
vertical nudging (WRF_deact_ndg), and nudging applied only above the PBL (WRF_ndg_abvPBL), allowing the lower
atmosphere to evolve freely. Notably, the original model configuration was preserved, with modifications restricted to the
nudging parameters. These experiments enabled us to evaluate how vertical nudging influences the simulation of the
cyclone, with a particular focus on its development and intensity.
Additionally, spectral nudging was applied to wavelengths longer than approximately 1,000 km to reduce distortions in the
large-scale circulation within the regional model domain. This approach helped maintain consistency between the model's
solution and its boundary conditions (Miguez-Macho et al., 2004). Model outputs were generated every 6 hours.
The verification of simulations is conducted by evaluating them against ERA5 reanalysis data and the Multi-Source
Weighted-Ensemble Precipitation (MSWEP; Beck et al., 2019).

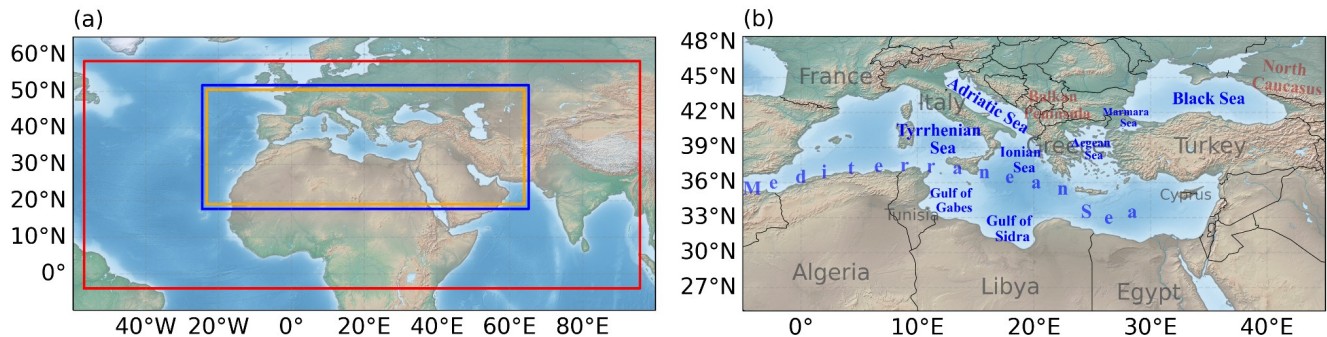

**Figure 1: (a) WRF and FLEXPART-WRF simulation domains. The WRF model includes two nested domains: d1 (18 km**
**resolution, red box) and d2 (6 km resolution, blue box). The FLEXPART-WRF domain (6 km resolution) is shown in yellow. (b)**
**Mediterranean basin.**
(b)  FLEXPART-WRF
In this study, we employed the Lagrangian FLEXPART-WRF model version 3.3.2 (Brioude et al., 2013), a modified version
of the FLEXPART Lagrangian particle dispersion model (Stohl et al., 2005) adapted to be fed with WRF output.
FLEXPART-WRF uses mesoscale model outputs to initiate the dispersion of active moisture tracers (or particles) in a
Lagrangian framework. The FLEXPART-WRF model was driven by the WRF outputs with a time step of 6 h. For the WRF
configuration described, the FLEXPART-WRF model was run at a spatial resolution of 6 km (Fig. 1, yellow box). In the
simulations, the atmosphere was partitioned into approximately 10 million air parcels distributed across the model domain.
The selection of particle numbers ensured a balanced distribution across both grid points and vertical levels. These parcels
were advected over time using the three-dimensional wind field from WRF.
The FLEXPART-WRF simulations incorporated an activated subgrid terrain effect parametrization and included convection
schemes throughout the simulation (Supplementary Material in Brioude et al., 2013). Vertical motion was represented
through vertical velocity, calculated from divergence. The model also included a turbulence scheme to account for turbulent
effects, with the Hanna scheme being used to specifically handle turbulence in the presence of activated convection (Hanna,



1984). This scheme accounts for boundary layer parameters, such as PBL height, Monin-Obukhov length, convective
velocity scale, roughness length, and friction velocity.

## 2.2 Ianos Footprints

Storm Ianos was detected and tracked using the mean sea level pressure (MSLP) field from WRF outputs. Initially, a pre-
filtering step was applied to identify potential cyclone centres within the MSLP field. This was achieved by identifying
locations where the MSLP anomaly—defined as the difference between the current MSLP and the 10-day running average—
fell below -3 hPa, in line with the method used by Bacmeister et al. (2014). Subsequently, candidate centres were selected
based on the criterion of representing a local minimum compared to the eight surrounding grid points. To track the cyclones
over time, each detected centre was matched to the nearest recognized cyclone centre within a 400 km radius six hours later.
In cases where multiple centres were identified within this critical distance, the centre with the lowest MSLP was selected as
the cyclone centre for the subsequent time step. Additionally, following Cavicchia et al. (2014), tracks with lengths shorter
than 100 km were excluded to eliminate persistent stationary lows. The cyclone phase space method (Evans and Hart, 2003)
was employed to filter the tracked low-pressure centres. This tracking method was applied between 12 and 24 September,
during which a single cyclone was consistent with the previously recorded Ianos storm track and lifespan, as determined
from reanalysis, satellite, and weather station data (Lagouvardos et al., 2021).. The method is codified in the tool CyTRACK
(Pérez-Alarcón et al., 2024).
The target region for capturing precipitation associated with medicane Ianos was defined as the storm's radius at each time
step, following the methodology of Rudeva and Gulev (2007). This approach is based on identifying the contour of the last
closed isobar. Figure 2 presents a representation of this methodology for a specific stage of Ianos' life on 17 September 2020
at 00 UTC.



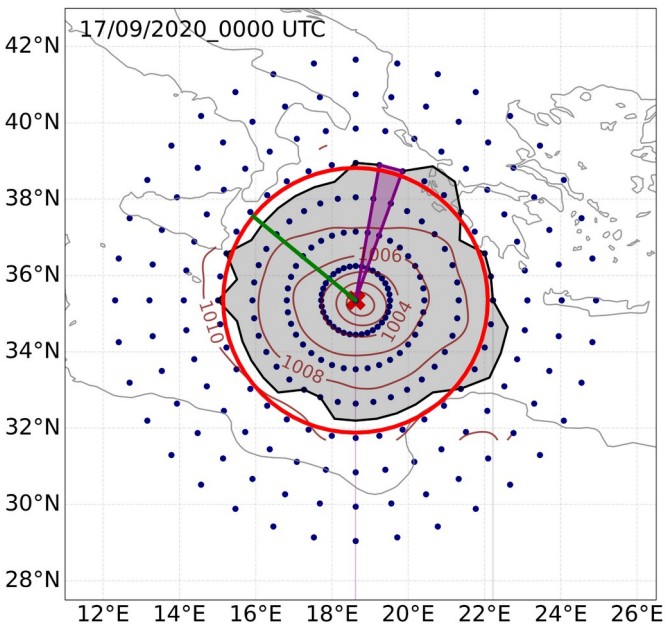


**Figure 2: Schematic representation of cyclone outer radius estimation based on the method developed by Rudeva and Gulev (2007). The method is applied to the MSLP field (contoured in the background) for Medicane Ianos on 17 September 2020 at 00:00 UTC, with a central position at 18.7°E, 35.5°N. Blue dots represent radial legs with 10 angular steps centered on the cyclone. The last closed isobar contour (bold black) encloses the cyclone area (shaded), which is determined by summing the areas of triangular sections formed between two consecutive points on the last closed isobar and the cyclone center (example shown in purple). The red circumference represents the equivalent circular area of the cyclone, whose radius is considered the cyclone's effective radius (in green).**

A pattern consisting of 36 radial legs was constructed, all originating from the low-pressure system's center (Fig. 2, dotted blue). These legs extended 1,000 km outward, spaced at 10-degree intervals. For each radial leg, the MSLP value where the first derivative of the pressure field approached zero was identified, marking the location of the last closed isobar in that specific direction. The minimum value among the 36 critical MSLP values was then interpolated for each radial leg, defining the contour of the last closed isobar (Fig. 2, black contour). The points along the last closed isobar and the centre of the cyclone formed triangles with known areas $A_n$. By summing the areas of triangles, we calculated the total area of the geometry under the last closed isobar (Fig. 2, shaded area):

$$A_G = \sum_{n=1}^{36} A_n \,, \tag{1}$$

The obtained area ($A_G$) was attributed to a virtual circumference, whose radius ($r_c$):

$$r_c = \sqrt[2]{\frac{A_G}{\pi}} \,, \tag{2}$$



was taken as the effective radius of the cyclone.

**2.3 Methods for Attributing Moisture Sources**

After determining the track and radius of Storm Ianos from the WRF model outputs and obtaining FLEXPART-WRF
outputs, the analysis focuses on air parcels, so-called particles, located inside the vertical atmospheric column within the
specified storm region. For each parcel, the water budget is computed to enable a precise attribution of moisture sources.
The Lagrangian method developed by Stohl and James (2004, 2005) identifies moisture sources and sinks by tracking the
trajectories and moisture content of individual air parcels over time. This approach tracks variations in specific humidity (q)
for each parcel at 6 hours intervals. By quantifying the changes in specific humidity along the trajectories of individual air
parcels, it becomes feasible to estimate the surface freshwater flux over a specified area. This is achieved by summing the
moisture contributions of all air parcels passing through a specific grid cell at a given time.  The total moisture budget in
each grid cell can then be expressed as:
$$E - P = \sum_{i=1}^{N} \Delta q_i,$$    (3)
where N is the total number of air parcels. The resulting evaporation minus precipitation ($E - P$) fields yield a cumulative
representation of moisture sources and sinks over the specified period.
For a backward trajectory, regions where evaporation exceeds precipitation ($E - P > 0$) are identified as moisture source
areas for the target region. These areas are characterized by a net atmospheric moisture release, where the rate of evaporation
surpasses that of precipitation, contributing moisture to the airmass as it moves toward the target location.
Upon the methodology by Sothl and James (2004, 2005), Sodemann et al. (2008) developed a method for quantifying
moisture sources for precipitation that involves backtracking air parcels while accounting for moisture gains and losses due
to precipitation. In this method, air parcels are selected based on whether they precipitate over the target region. Precipitating
particles are identified as those whose specific humidity decreases by more than 0.1 g/kg before reaching the target region
(Läderach and Sodemann, 2016). The contributions of moisture uptake along the parcel's trajectory are weighted according
to the initial specific humidity values. If specific humidity decreases, the moisture uptake is proportionally adjusted to
account for the loss of moisture due to precipitation during transport. No distinction is made for moisture uptake within and
above the boundary layer (Fremme and Sodemann, 2019; Sodemann, 2020).
Sothl and James (2004, 2005) and Sodemann et al. (2008) approaches were implemented in the Lagrangian TROVA tool
(Fernández-Alvarez et al., 2022). In this study, particle trajectories were tracked over 10-day intervals, a typical time frame
commonly used to trace atmospheric water vapour in studies of moisture source-sink dynamics (Numaguti, 1999; van der
Ent and Tuinenburg, 2017; Gimeno et al., 2021). Subsequently, the methodology of Sodemann et al. (2008) was applied.



## 4 Results and Discussion

### 4.1 Storm Ianos's Track and Structure

The impact of different weather prediction model configurations on the genesis and subsequent track of Storm Ianos has been thoroughly investigated (Ferrarin et al., 2023; Pantillon et al., 2024; Sanchez et al., 2024). In this study, we evaluated the performance of the gridded nudging method as outlined in Section 2.1. While this is not intended as a full sensitivity analysis, our primary objective is to identify the configuration that best represents the cyclone's track and structure to ensure the highest quality for subsequent moisture source analyses.

It is worth noting that our WRF experiment design begins several days before the initial low-pressure disturbance rather than closer to its onset, as done in previous studies. This choice is due to the requirements of the moisture source attribution methodology, which relies on backward particle trajectory analysis. This approach requires meteorological data from WRF outputs for up to 10 days.

Figure 3a compares the cyclone tracks derived from ERA5 reanalysis data and the three WRF model configurations (Section 2.1a). The genesis phase exhibits the greatest sensitivity and variability in the simulated trajectories. The warm waters of the Gulf of Sidra serve as the genesis location for Ianos, following a series of prior convective storms (Lagouvardos et al., 2022). The configuration with gridded nudging above the PBL effectively captures the system's formation region and early evolution, showing improved alignment with the ERA5 reference track. The trajectories over the Mediterranean deviate from the reanalysis track in all WRF configurations. However, as the system moves over the Ionian Sea and Greece, the trajectories align more closely, with the configuration using nudging above the PBL showing the best agreement with the ERA5 track.

Figure 3b illustrates the evolution of the minimum central MSLP throughout Ianos' lifecycle for ERA5 and the three WRF configurations. Among them, the MSLP time series from the WRF configuration in which the PBL evolves freely shows the closest agreement with ERA5, although it still represents a more intense cyclone. The meteorological stations in Palliki (Cephalonia), Skinari (Zakynthos), and Pylos Buoy collected data on the passage of Ianos. Notably, detailed information on the core of the storm was recorded in Palliki. On 18 September, at 0500 UTC, a minimum MSLP of 984.3 hPa was recorded in Cephalonia, while Zakynthos recorded a minimum of 989.1 hPa (Lagouvardos et al., 2022). The minimum MSLP of Ianos in the simulations occurs approximately on day 18 at 0600 UTC (Fig. 3b). Despite the time lag, the MSLP profile of Ianos in the simulation, using the configuration with nudging above the PBL, closely matches the MSLP observations recorded in Palliki and Skinari during the storm's influence. Additionally, the cyclone center location from that track remains closer to the distances observed for ERA5 relative to the reference stations (Fig. 3b). On 19 September, at 0300 UTC, the pressure data from the Pylos buoy, although located farther from the cyclone center (76.7 km from the ERA5 track), still captured a decrease in MSLP, peaking at 1008 hPa, which further corroborates the observed profiles.





An evaluation of Ianos's tropical transition was conducted by comparing phase space diagram parameters, including upper-
and lower-level thermal winds and thickness symmetry (Supplementary Material Section A). The simulation with nudging
above the PBL was the most accurate in representing the vortex's tropicalization, as reflected in the evolution of thermal
winds (Fig. S1). During the first 24 hours, all WRF configurations exhibited a symmetric core, whereas ERA5 data indicated
a frontal system. However, beyond this period, the simulations aligned more closely with the reanalysis data. We tested the
radius used to estimate these phase parameters (Fig. S2). We found that the values of the phase parameters remained within
the predefined thresholds that determine each classification: warm or cold core, symmetric or asymmetric.

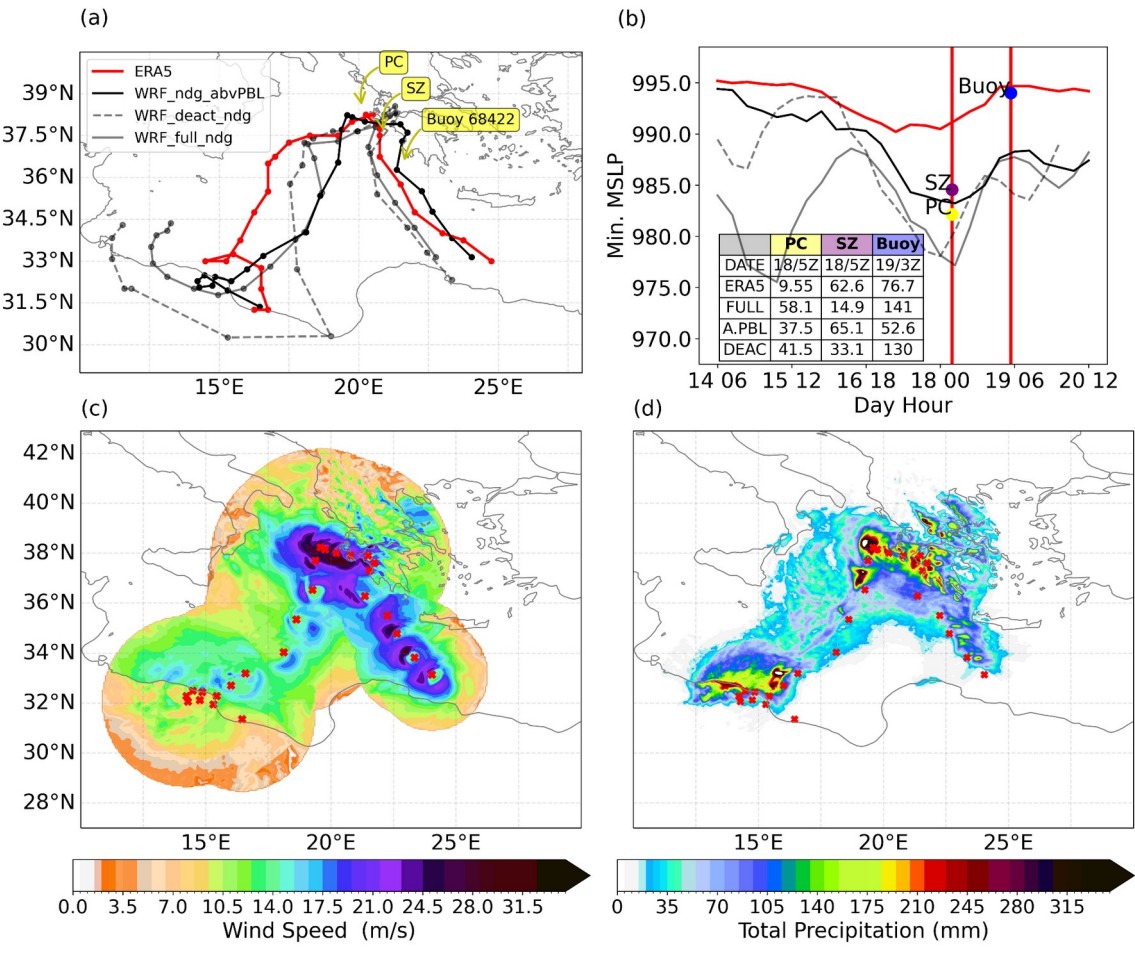


**Figure 3: (a) Tracks of Storm Ianos derived from ERA5 reanalysis data and WRF configurations, spanning from 14 September**
**2020, at 0600 UTC to 20 September 2020, at 1200 UTC, with 6h intervals. The map highlights the locations of surface**
**meteorological stations: Palliki (Cephalonia)-PC, Skinari (Zakynthos)-SZ, and the Pylos buoy. (b) Minimum MSLP evolution.**
**Meteorological stations recorded the minimum MSLP associated with the passage of Ianos on 18 September at 0500 UTC and 19**
**September at 0300 UTC (scattered). The table presents the distances between the cyclone center and the stations at the closest time**
**steps to the recorded data. Based on WRF model outputs with nudging applied above the PBL: (c) Maximum surface wind speed,**
**and (d) Accumulated precipitation, both at 6h intervals.**



Figures 3c and 3d show maximum surface winds and 6 hours accumulated precipitation for the above PBL nudging
configuration, which is used hereafter due to its superior alignment with the ERA5 reference track and additional verification
tests. These tests involve the evaluation of model outputs against ERA5 reanalysis data for wind, temperature, and specific
humidity across the lower, middle, and upper atmosphere. Additionally, spatial verification of precipitation is conducted
using the MSWEP dataset (Supplementary Material, Section B).
The wind field accurately represents the intensification of winds as the storm approaches the southern Ionian Sea, ranging
from above 20 m/s to 25 m/s around the centre, as recorded by satellite observations (Ferrarin et al., 2023). A peak in
accumulated precipitation is simulated both at this step and as Ianos approaches landfall (Fig. 3d). Another precipitation
peak is seen in the early stages, associated with the convective cluster in the Gulf of Sidra, when Ianos was in its initial phase
(Fig. 3d). Of particular interest is the precipitation peak observed as Ianos approaches Greece, where the effective
precipitable water amounts were unusually high, comparable to those seen in tropical regions, as described by Lagouvardos
et al. (2022).
Storm Ianos reached this stage through the interaction of upper-tropospheric precursors with their low-level counterparts.
Accurately representing the tropospheric structure, particularly the evolution of medicane dynamics, is essential. Figure 4
shows the evolution of the upper-level structures (at 300 hPa) and Figure 5 illustrates cross-sections of meteorological fields
from WRF outputs. At upper levels, a trough extended over the Mediterranean, positioning the disturbance beneath a jet
stream between 30° and 32°N is simulated during the early stages of Ianos on 14 September at 0600 UTC (Fig. 4a). A
notably distinctive feature of the Ianos case is the development of a region between 32° and 40°N characterized by
tropospheric PV values (PV < 2 PVU), referred to as the "low-PV values bubble," in the subsequent step (Fig. 4b). This
phenomenon, first described by Sanchez et al. (2024), is linked to the influence of the preceding cluster of convective storms
and is also apparent in the vertical cross-section at 0600 UTC on 15 September due to the tropopause fold (Fig. 5b). Other
characteristics commonly observed in medicanes were identified, such as the stratospheric PV streamer that eventually
undergoes cyclonic wrapping around Ianos's surface location (Figs. 4c to 4f). This feature typically occurs in the phases
leading up to the intensity maximum in major medicanes (Flaounas et al., 2021) and during the axisymmetry tropical
transition process (Miglietta et al., 2011).



**Figure 4: Storm Ianos evolution (surface centre marked in red) from genesis to mature phase. Depicted at 300 hPa level: potential vorticity (shaded), geopotential height (contours in kilometres), and wind vectors (barbs).**

Additionally, Fig. 5 illustrates the completion of the vertical coupling between the stratospheric intrusion and the positive PV in the lower and middle troposphere, the PV tower (Wernli and Davies, 1997). The positive PV anomaly extends into the troposphere, along with the symmetry of the equivalent potential temperature (Fig. 5e and 5f), causing the system's profile to resemble that of tropical cyclones (Thorpe, 1985; Galarneau et al., 2015). On 18 September at 0600 UTC (Fig. 5e), the relative humidity at 80% shows a maximum on either side of the centre, with a minimum in between. This pattern corresponds to reduced convection at the "eye" of the system, where precipitation becomes more concentrated around it.





**Figure 5: South-North cross-sections along the central longitude of Storm Ianos, with the position indicated by the dashed line, from genesis to the mature phase. Shaded potential vorticity, red contours of equivalent potential temperature, 20% relative humidity in the solid blue line, and 80% relative humidity in the dashed blue line.**




In this context, the following section analyzes moisture transport and the major source regions involved during the intensification and
mature stages of Ianos, from September 16 to September 19, coinciding with the record-breaking rainfall.

## 4.1 Moisture Transport and Water Budget Associated with Ianos

Figure 6a illustrates the integrated moisture uptake field for the Storm Ianos pathway, from its genesis on 14 September to its lysis on 20
September 2019. The highest moisture contributions are concentrated in the Ionian Basin and the southwestern Balkan Peninsula, regions
that were directly affected by the cyclone's trajectory. Lower moisture contributions, though still aligned with the storm's path, extend
beyond its direct influence, particularly over areas in North Africa, including Libya and Tunisia, as well as the Tyrrhenian Basin and
southern Italy. Additionally, the Aegean, Marmara, and Black Seas serve as secondary moisture source regions, contributing to the overall
moisture uptake associated with the storm.

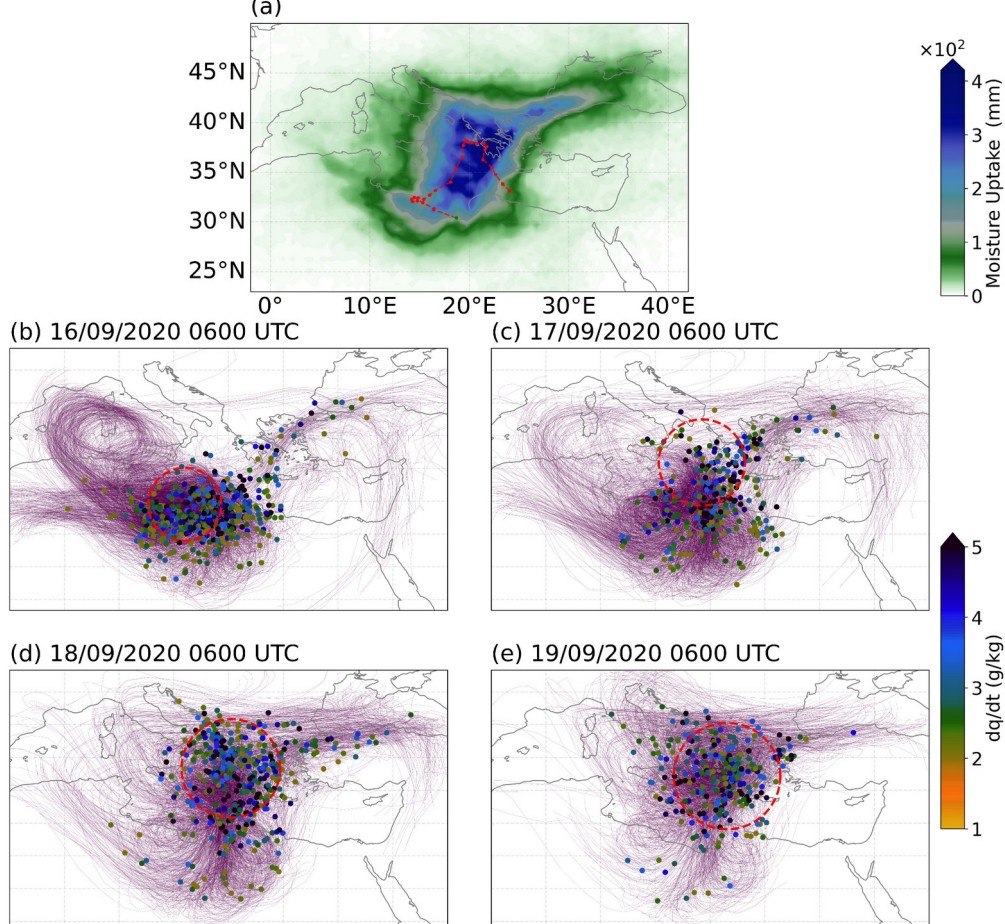

**Figure 6: (a) Integrated moisture uptake field along the full pathway of Storm Ianos (indicated by the red dotted line), from its**
**genesis on 14 September at 0000 UTC to its lysis on 20 September 2019 at 0600 UTC. (b-e) Trajectories associated with cyclone**



**precipitation during the intensification and mature stages of Ianos, from September 16 to September 19. Also shown are 6-hour**

**changes in specific humidity, with increases exceeding 1.5 g/kg.**

Figures 6b to 6e depict a selection (for visibility purposes) of pathways of precipitating particles at different stages of Storm Ianos. Locations where specific humidity increases by more than 1.5 g/kg are highlighted to emphasise regions of notable moisture gain. On 16 September 2020 at 0600 UTC (Fig. 6b), the centre of Storm Ianos exerted its influence over the Gulf of Sidra. Precipitating particles in this region originate from three primary locations: i) from the Black Sea, crossing the Marmara Sea, then moving between Greece and the Dodecanese Islands, ii) from the Tyrrhenian Basin, following a circulation pattern that carries them over the Algerian Basin, then southeastward across Libya before entering the target region from the south, and iii) from Northwest Africa, following an approximately zonal eastward path that brings them across the Gulf of Gabes. Pathways from the Tyrrhenian Basin and Northwest Africa align with the atmospheric flux driven by the high- to mid-level synoptic circulations (Fig. 4), characterized by a quasi-horizontal flow from the west and a low-pressure system to the west of Italy. The remaining pathways are likely influenced by lower-level circulations and orographic effects, which will be discussed later.

Between 17 and 19 September (Figs. 6c, 6d, 6e), the storm approached Greece, while atmospheric precipitating particles continued to concentrate along pathways extending over North Africa, Libya, the Gulf of Sidra, the Ionian Sea, and the Black Sea. Specific humidity increases primarily occurred near the storm centre and along the routes stretching from the northeastern region, through the Marmara Region, and down to the Tunisia-Libya coastal area.





**Figure 7: Three-dimensional distribution of particles with specific humidity decreasing by more than 0.1 g/kg in 6 hours before reaching the radius of Storm Ianos at the following times: (a) September 16, 0600 UTC, (b) September 17, 0600 UTC, (c) September 18, 0600 UTC, and (d) September 19, 0600 UTC. The distribution of particles is projected onto the storm's radius (x, y). The x-axis represents latitude, the y-axis represents longitude, and the z-axis represents the vertical distribution of particles within 100 hPa layer intervals, represented by color. Panels (e) to (h) show the moisture uptake field for core precipitation at each corresponding time step. The radius of the storm is represented by the red dashed line, with the centre of the system marked by the red cross. Panels (i) to (l) illustrate the vertical cross-section of moisture uptake along the Marmara region for each step, along the purple line in (e) to (h).**



Figures 7a to 7d depict the distribution of particles with specific humidity decreasing by more than 0.1 g/kg in 6 hours before reaching the
radius of Storm Ianos at the corresponding stages shown in Fig. 6. Specific humidity in an air parcel is influenced not only by evaporation
and precipitation but also by microphysical cloud processes, phase changes, and atmospheric mixing (Sodemann, 2020). Within a
medicane's radius, cloud microphysics, atmospheric mixing, and convective transport critically redistribute moisture. This is reflected in
our detection criteria results (Figures 7a–7d). As the system approached its peak intensity (Figs. 7c and 7d), there was a notable increase in
the number of precipitating particles at various atmospheric levels. This increase was especially pronounced between 900 and 600 hPa,
where the highest concentration of particles was observed. This trend is indicative of the intensification of the precipitation process within
the storm. Satellite measurements reveal the precipitation structure and microphysical processes of the medicane, confirming that the most
intense precipitation occurs below the freezing level, which ranges between 4 and 5 km (D'Adderio et al., 2022).
Furthermore, the distribution of precipitating particles corresponds with developing a more organised cyclonic circulation, which deepens
over time and extends to higher altitudes. For example, the comparison between Figures 7a and 7c illustrates this progression, showing a
shift in particle distribution from 800 to 400 hPa around the medicane centre and the distinct precipitation-free "eye" structure. Satellite
data further reveals the formation of a closed "eye" by September 16 at 13:40 UTC, characterised by a nearly circular area with little to no
rain within a 50 km radius of the cyclone's centre (D'Adderio et al., 2022). Additionally, Ianos' precipitation structure, as revealed by
satellite observations, signals exceptionally deep convection for the Mediterranean region. Suspended and heavily rimed ice particles are
observed in the outflow region surrounding the main convective cores (Hourngir et al., 2021; D'Adderio et al., 2022). Storm Ianos is
characterized by a well-developed vertical structure, as shown in Fig. 5. Although Figs. 7a to d show a lower particle density at higher
altitudes, their presence remains detectable, with particle density increasing as the cyclone intensifies. This observation suggests the
presence of a vertically extended mixed-phase region and strong updrafts.
The moisture sources that fueled these processes throughout the storm's development are depicted in Figs. 7e to 7h. Moisture is
predominantly uptaked in the vicinity of the medicane along its path. In the Central Mediterranean, source regions are identified in the
Adriatic Sea, Gulf of Sidra, and Ionian Sea. As the medicane progresses into this region, the intensity of moisture uptake increases,
revealing the evaporative strength of the cyclone. During landfall and the record-breaking rainfall in Greece, the Adriatic Sea emerged as
the primary source of moisture. Notably, moisture sources over the central Mediterranean have been highlighted as major contributors of
water vapor for high-rainfall-producing cyclones in the basin. The higher average sea surface temperature (SST) of cyclones producing
high rainfall, compared to those producing low rainfall, has been identified as a key factor driving the increased intensity of moisture
uptake over the Mediterranean (Flaounas et al., 2019). This trend is consistent with the positive SST anomalies observed over the
Mediterranean during the life of Ianos (Fig. S8).
The consistent moisture uptake from the Black Sea region is particularly noteworthy, as it represents the most significant remote source.
Figures 7i to 7l present vertical cross-sections of moisture uptake over this region. The cross-sections reveal that moisture is taken up from
the surface up to approximately 850 hPa. As the system intensifies, the moisture flux from this region progressively increases. A more
detailed analysis is presented, based on a step-by-step examination of moisture uptake.




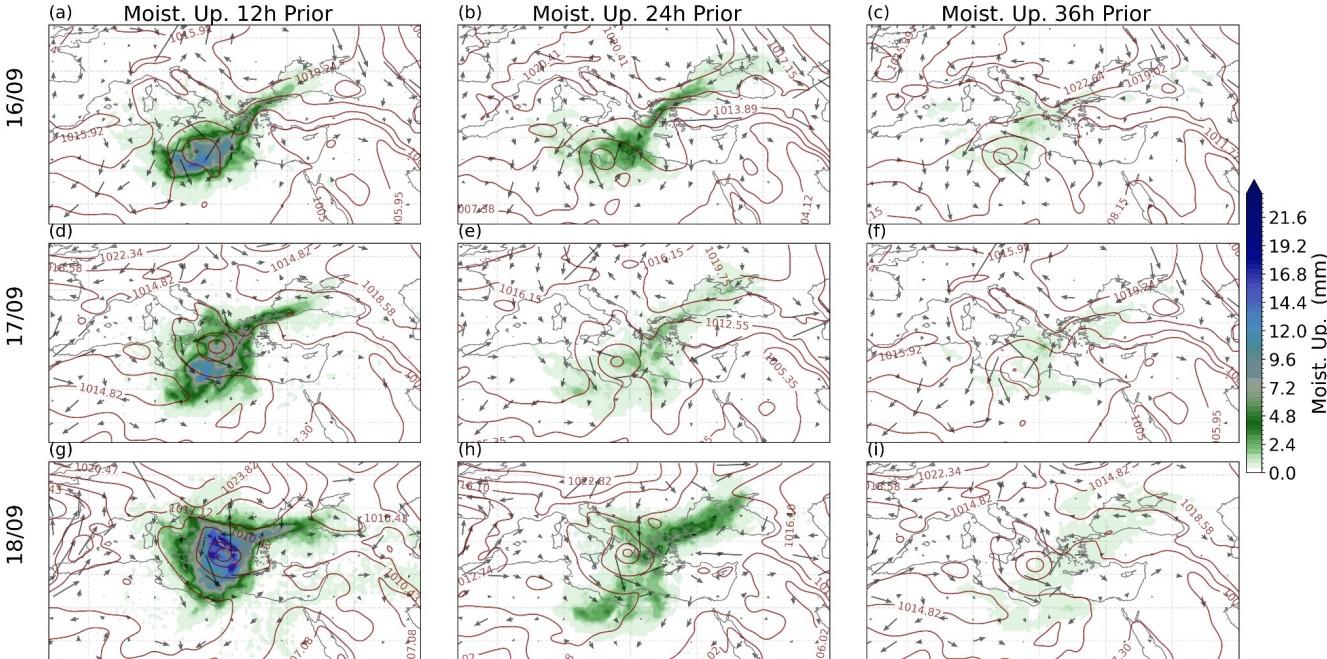

**Figure 8: 12 hourly accumulated moisture uptake over 36 backtracked hours for precipitating particles within the radius of Storm Ianos at 0600 UTC on September 16 (top), 17 (middle), and 18 (bottom). Red contours denote mean sea-level pressure (MSLP), and arrows represent vertically integrated moisture flux (VIMF) at 0600 UTC of each respective start date in 24-hour steps, derived from WRF.**

Figure 8 presents the daily moisture uptake during the final 36 hours before the particles reach their target. For North Atlantic extratropical cyclones, it has been shown that the majority of moisture uptake happens in the short period just before precipitation, with uptake occurring within 2 days of rainfall contributing to more than 50% of the total precipitation (Papritz et al., 2021). Daily values in Fig. 8 represent daily contributions cumulatively forming a subset of the total moisture uptake shown in Figs. 7e to 7h. The daily MSLP and vertically integrated moisture flux (VIMF) fields at 0600 UTC, depicted in Fig. 8, provide a contextual framework for understanding the drivers of moisture transport.

The synoptic background reveals three primary modulators of moisture flux over the domain: a high-pressure centre moving through southeastern Europe, a northwest-to-southeast flow over the North Caucasus region influenced by a retreating low-pressure area to the north, and a cyclonic circulation over the eastern Black Sea (e.g. Fig. 8a). The eastern ridge of high pressures imposes a northeast-southwest flow over the Black Sea, directly towards the Mediterranean, with its orientation fluctuating depending on the interaction with the cyclonic circulations to the east. An oriented flow through the Black Sea to the Ionian Sea occurs between September 14 (Fig. 8c, also shown in Fig. 8f) and 16 (Fig. 8a, also shown in Figs. 8e,i), which subsequently weakens due to the retreat of the southeastern European high-pressure system and the low-pressure systems towards the east.

The regions that contribute most intensively to this moisture uptake in the vicinity of the storm are observed in the final 24 hours, highlighting the in-situ evaporative processes associated with the storm. However, moisture from the Black Sea region is absorbed within a timeframe of 48 to 72 hours before reaching the target area. The synoptic conditions in this region reveal an intensification of pressure




gradients, which create favorable conditions for enhanced evaporation due to stronger winds. These winds are further channeled by the
high orography of the Balkans and Turkey, amplifying the moisture uptake. Previous studies on Mediterranean cyclones have shown that a
low-level jet over the Black Sea plays a significant role in influencing cyclone development (Raveh-Rubin et al., 2016).
The Libya- Tunisia region also serves as a notable moisture source, as indicated by the moisture uptake observed in Fig. 6. During Storm
Ianos's lifespan, this region exhibited exceptionally high levels of available water vapor (Fig. S9). Before Ianos, a convective event in the
Gulf of Sidra enhanced atmospheric water availability, with southerly surface winds subsequently transporting this moisture inland (Figs.
S9a to c). As Ianos moved away from the coast, precipitation efficiency decreased (Fig. S10). However, the contribution of southerly
winds and the availability of atmospheric moisture (Fig. S11), particularly in the lower levels, remained substantial (Fig. S12). During the
period analyzed in Fig. 8, the primary moisture uptake occurred between September 15 and 17.
Figure 9 depicts the 6-hourly instantaneous moisture uptake for Ianos on September 17, 0600 UTC (corresponding timeframe in Fig. 8d).
In the first 6 hours of the backward trajectories (Fig. 9a), moisture is primarily gained over the Mediterranean Sea. For the 12-hour
backward trajectories, moisture is gained over a region in inland Libya, with a more intense uptake, likely due to the higher particle density
in this region before continuing toward the target area. Less moisture is gained 18 hours prior, as the number of particles in the area,
though still within a moist environment, is lower compared to the previous step (Fig. S12).

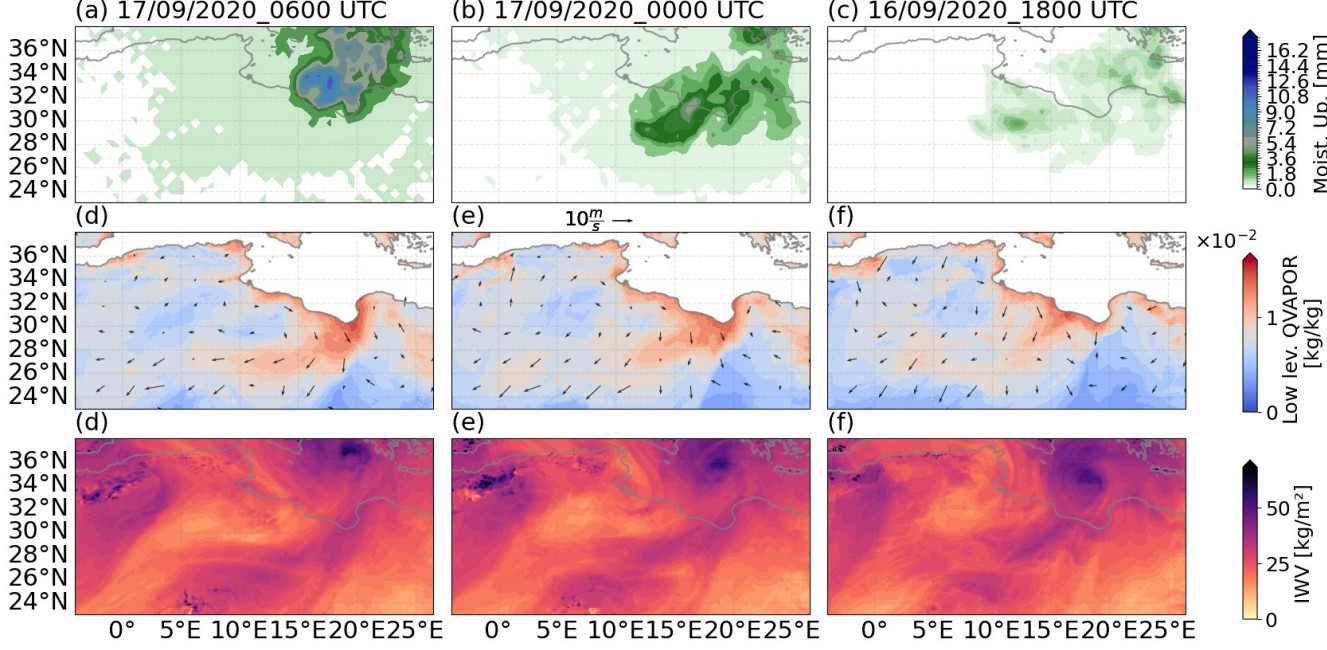


**Figure 9: (a–c) The 6-hourly instantaneous moisture uptake along the backward trajectories of precipitating particles for Ianos,**
**starting on September 17, 0600 UTC, corresponding to the timeframe in Fig. 8d. (d–f) Mean Water Vapor Mixing Ratio**
**(QVAPOR) averaged between 1000 and 750 hPa, with surface winds represented by arrows. (g–i) Integrated Water Vapor**
**Column (kg/m²).**

387

35                                                                    18
36





## 5 Concluding Remarks

Storm Ianos, a Mediterranean medicane, has attracted considerable attention from the scientific community. This unprecedented case challenges established paradigms, raising critical questions about its dynamics and impacts. Among these, a key question arises: What fueled the record-breaking rainfall of Storm Ianos? In this study, we employed a Lagrangian approach to identify and attribute the moisture sources of precipitating air particles within the radius of Ianos.

Our methodology involves tracking atmospheric particles within a complex and dynamic environment, as revealed by high-resolution weather simulations. The cluster of convective storms preceding the genesis of Ianos acted as a precursor to the baroclinic conditions that later facilitated its intensification. This evolution was marked, for instance, by the development of a low-PV bubble at upper levels, interacting with a stratospheric PV streamer (Fig. 4 and 5; see also Sanchez et al., 2024). Another notable mesoscale feature resolved in the simulations is the development of an axisymmetric warm core (see Supplementary Material, Section A). Changes in relative humidity and equivalent temperature levels further signal this transition. Additionally, a PV tower forms, reflecting the intensification of diabatic processes beneath the stratospheric intrusion. The improved cyclonic organization, the deepening of precipitation processes, and the structure of precipitation are captured in the representation of precipitating parcels derived from dispersion model outputs within the cyclone's tropospheric column during its intensification to maturity (Fig. 6). Signs of less-developed convection within the cyclone's "eye" are also displayed.

The moisture contributing to precipitation during Ianos's lifecycle was distributed across the Mediterranean, with primary sources located near its trajectory over the Ionian Basin and the southwestern Balkan Peninsula. Additionally, more distant contributions were traced to the Black Sea. During the intensification and mature stages, analyses confirmed that the fractions of moisture gains directly contributing to precipitation within the target region were higher along the trajectories of particles in proximity to the target region. Notably, the attribution methodology accounts for precipitation losses, thereby highlighting the representativeness of local moisture gains.

Precipitating particles originate from three primary pathways: from Northwest Africa along a zonal eastward path crossing the Gulf of Gabes; from the Black Sea via the Marmara Sea and between Greece and the Dodecanese Islands; and from the Tyrrhenian Basin through the Algerian Basin and Libya before entering from the south, with the latter two transport pathways persisting during the intensification and mature stages. This is consistent with the persistence of source regions over the Libya-Tunisia region, the Gulf of Sidra, the Ionian Sea, and the Marmara-Black Sea area.

An analysis of moisture uptake accumulated over 12-hour intervals during the 36 hours preceding precipitation reveals that the most substantial gains occur within the final 24 hours, primarily in the vicinity of the storm's radius. This is mostly linked to the storm's increased evaporative power and the intensification of its winds during this period. Remote sources are most prominent between 12 and 24 hours before precipitation. The primary remote source is the Black Sea, where moisture uptake occurs below 850 hPa, which intensifies as the system develops. The transport mechanism supporting this source is influenced by atmospheric circulation patterns that channel moisture from the Black Sea region. These patterns are driven by



several factors, including a high-pressure system moving through southeastern Europe, a northwest-to-southeast flow over
the North Caucasus region associated with a retreating low-pressure area to the north, and a cyclonic circulation over the
eastern Black Sea. These combined circulation dynamics facilitate the advection of moisture towards the storm, enhancing
its moisture supply.
This study identifies the moisture sources associated with a complex cyclonic system. To address the limitations inherent in
its dynamic evolution, high-resolution numerical simulations were utilized. These simulations successfully captured critical
moments characterizing the exclusive environmental conditions underlying this unique case. The moisture tracking results
revealed new questions regarding the origins of the moisture, highlighting areas for further investigation. Future
investigations could focus on the variability of moisture sources across different medicanes and assess how regional and
remote contributions shift under varying atmospheric conditions. Advancing understanding in this domain holds significant
implications for the broader scientific community.
**Code availability**
The        WRF        and        WRF-FLEXPART        models        can        be        downloaded        from
https://www2.mmm.ucar.edu/wrf/users/download/get_source.html        and        https://www.flexpart.eu/wiki/FpRoadmap,
respectively.    The    TROVA    codes    and    executables    are    stored    in    the    permanent    link
https://github.com/tramo-ephyslab/TROVA-master. CyTRACK can be found at https://github.com/apalarcon/CyTRACK.
**Data availability**
The    ERA5    reanalysis    dataset    was    freely    obtained    from    the    Copernicus    Climate    Data    Store
(https://cds.climate.copernicus.eu/cdsapp#!/search?type=dataset). The MSWEP global precipitation product is available
upon request from GloH2O (https://www.gloh2o.org/mswep/). Pylos buoy data were accessed through the POSEIDON
network,    operated    by    the    Hellenic    Centre    for    Marine    Research,    and    are    also    available    upon    request
(https://poseidon.hcmr.gr/).
**Author contribution:**
PC-H: Conceptualization,    Methodology,    Investigation,    Software,    Visualization,    Writing—original    draft.    RN
Methodology, Supervision, Writing—review and editing. AR   Conceptualization, Methodology, Validation, Writing—
review and editing. PL,  Conceptualization,  Software, Validation, Writing—review and editing. LG Conceptualization,
Methodology, Supervision, Writing—review and editing. RN and LG, Funding acquisition.





**Competing interests:**

The authors declare that they have no conflict of interest.

**Acknowledgements:**

The authors would like to thank Prof. Joaquim Pinto for his valuable input and insightful discussions during the early stages of this work. P.C.-H. acknowledge the support from the Xunta de Galicia (Galician Regional Government), the «Programa de axudas á etapa pre doutoral da Xunta de Galicia Galicia (Consellería de Cultura, Educación, Formación Profesional e Universidades)» under grant no. ED481A-2022/128. Furthermore, EPhysLab members are supported by the SETESTRELO project (grant no. PID2021-122314OB-I00) funded by the Ministerio de Ciencia, Innovación y Universidades, Spain (MICIU/AEI/10.13039/501100011033), Xunta de Galicia under the Project ED431C2021/44 (Programa de Consolidación e Estructuración de Unidades de Investigación Competitivas (Grupos de Referencia Competitiva) and Consellería de Cultura, Educación e Universidade), and by the European Union 'ERDF A way of making Europe' "NextGenerationEU"/PRTR. This work has also been possible thanks to the computing resources and technical support provided by CESGA (Centro de Supercomputación de Galicia) and RES (Red Española de Supercomputación).

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
