# Peer review of "Lagrangian Tracking of Moisture Sources for the Record-Breaking 2 Rainfall of Storm Ianos"

_EGUsphere, 2025_

## Author Comment (AC1)

**Response to Reviewer #1**

We would like to express our gratitude to the reviewer for their insightful and constructive feedback. In response to the observations of the reviewer, we have initiated simulations for two other Ionian cyclones, Zorbas and Daniel, using the same methodological framework as employed in the current study. Because the identification of moisture sources relies on backward-time Lagrangian analysis, these new simulations cover extended periods, resulting in significantly increased computational demands. Furthermore, we are conducting a sensitivity test using higher temporal resolution, which we believe could offer added value to the analysis, should the reviewer deem it relevant to the scope of the present manuscript. We respectfully request that the editor consider allowing time for the completion of these additional simulations and the incorporation of their results.

We have made every effort to address the reviewer's comments thoroughly. Reviewer comments are presented in bold, followed by our responses.

**MAJOR POINTS:**

1. **...The moisture uptake was discussed for medicane Qendresa in a previous paper by some of the authors (Coll-Hidalgo et al., 2023), sharing some points with the present paper; also, the paper has some analogies with the work done in Scherrman et al. (2023) and Miglietta et al. (2021), although they consider a different perspective for a similar purpose. A comparison with these works would probably highlight some peculiarities of this case and some analogies with the results in these studies.**

   We agree that a more explicit comparison with the studies by Coll-Hidalgo et al. (2023), Scherrman et al. (2023), and Miglietta et al. (2021) would enrich the discussion and better highlight both the peculiarities and the analogies of the present case study. In the revised manuscript, we will incorporate a more detailed comparative discussion with these studies.

2. **...Ianos has already been extensively studied in the scientific literature... I think some generalizations of your results to other cyclones is required, for example applying the same methodology (shortly) to other Ionian cyclones (e.g., Zorbas or Daniel) and drawing some general conclusions.**

   We agree that the case of Ianos has received considerable attention in the literature. Our objective, however, was to offer a novel Lagrangian-based quantification of moisture sources and uptake timing, employing high-resolution WRF simulations coupled with FLEXPART, which allow us to resolve near-core moisture uptake processes during the final intensification phase of Ianos.

   We fully acknowledge the reviewer's suggestion to strengthen the study by comparing Ianos with other similar medicanes. Following this recommendation, we have initiated the simulation and analysis process for two additional Ionian medicanes: Zorbas (2018) and Daniel (2023), which share geographical and seasonal similarities but exhibit different structural and dynamical evolutions.

   As of the date of this response, the simulations and Lagrangian tracking for these cases are in progress, but not yet complete. With the reviewer's agreement and the editor's approval

for additional time, we will finalize this comparative component and integrate it into the revised manuscript.

3. **Are 6 hours enough to represent correctly the moisture uptake? A comparison with the results obtained using a finer temporal resolution (1 h?), at least for part of the trajectory, would convince me that the procedure is appropriate.**

We agree that temporal resolution can influence the identification of moisture source regions and uptake timing, particularly in fast-evolving systems such as medicanes. At present, our priority is to complete the simulations for the additional cases of Zorbas (2018) and Daniel (2023), which form part of the expanded comparative analysis. However, contingent on the editor's approval of an extension to the revision timeline, we plan to conduct a dedicated sensitivity experiment using 1-hour output to directly assess the robustness of our results against the current 6-hour baseline.

We would like to emphasize that our use of a 6-hour time step is informed by established precedent in the literature. Previous studies employing similar Lagrangian diagnostic methods (e.g., Läderach & Sodemann, 2016; Fremme & Sodemann, 2019) have shown that a 6-hour resolution provides a reasonable compromise: it captures the main physical processes while avoiding excessive numerical noise that may arise with shorter intervals. As discussed in Appendix B of Fremme & Sodemann (2023), reducing the time step can lead to more precise source attribution but may also amplify the impact of interpolation errors, particularly when the specific humidity threshold used to distinguish between evaporation and precipitation events approaches the magnitude of numerical fluctuations. Conversely, time steps longer than 6 hours tend to degrade trajectory fidelity and may obscure diurnal variability, ultimately reducing the diagnostic accuracy of source attribution. The estimated uncertainty in moisture source distance associated with time-step selection is on the order of 10–20%, or several hundred kilometers.

Nevertheless, we fully acknowledge the importance of verifying the sensitivity of our results to this choice. We intend to include the outcome of the planned 1-hour resolution test in the revised manuscript.

Fremme, A. and Sodemann, H.: The role of land and ocean evaporation on the variability of precipitation in the Yangtze River valley, Hydrol. Earth Syst. Sci., pp. 2525–2540, https://doi.org/10.5194/hess-23-2525-2019, 2019.

Fremme, A., Hezel, P. J., Seland, Ø., and Sodemann, H.: Model-simulated hydroclimate in the East Asian summer monsoon region during past and future climate: a pilot study with a moisture source perspective, Weather and Climate Dynamics, 4, 449–470, https://doi.org/10.5194/wcd-4-449-2023, 2023.

Laederach, A. and Sodemann, H.: A revised picture of the atmospheric residence time of water vapour, Geophys. Res. Letters, 43, 924–933, https://doi.org/10.1002/2015GL067449, 2016.

4. **Please clarify the need of the additional parameterizations in FLEXPART: is not the vertical wind field already contained in the WRF model outputs? what is the need of activating the convection and turbulence schemes?**

We appreciate the reviewer's insightful question regarding the need for additional parameterizations in FLEXPART-WRF. While the vertical wind component and the full three-dimensional wind field are indeed provided by the WRF outputs and used by FLEXPART for advective transport, the inclusion of turbulence and convection schemes in FLEXPART serves a critical complementary role.

Following recommendations from the FLEXPART-WRF framework (Brioude et al., 2013), without turbulence parameterization, FLEXPART operates as a non-dispersive Lagrangian trajectory model. Using the Hanna turbulence scheme (Hanna, 1982), FLEXPART internally computes planetary boundary layer (PBL) turbulent mixing based on WRF-derived parameters such as PBL height, Monin–Obukhov length, convective velocity scale, roughness length, and friction velocity.

Regarding convection, the choice to activate convective parameterizations depends primarily on the WRF model's horizontal grid spacing. Convective schemes are generally recommended for grid spacings larger than ~30 km, where convection is unresolved, while grid spacings finer than ~10 km typically allow convection to be resolved explicitly, especially below ~2 km. For intermediate resolutions, such as our ~6 km domain, no strict consensus exists; however, we adopted the Kain-Fritsch scheme in agreement with established literature (e.g., Miglietta et al., 2015; Fita and Flaounas, 2018; Miglietta et al., 2021). Consequently, consistent with Brioude et al. (2013), when a convective scheme is employed in WRF, the corresponding convective parameterization should also be activated in FLEXPART, to parameterize subscale convection.

Brioude, J., Arnold, D., Stohl, A., Cassiani, M., Morton, D., Seibert, P., Angevine, W., Evan, S., Dingwell, A., Fast, J. D., Easter, R. C., Pisso, I., Burkhart, J., and Wotawa, G.: The Lagrangian particle dispersion model FLEXPART-WRF version 3.1, Geosci. Model Dev., 6, 1889–1904, https://doi.org/10.5194/gmd-6-1889-2013, 2013.

Fita L, Flaounas E. Medicanes as subtropical cyclones: the December 2005 case from the perspective of surface pressure tendency diagnostics and atmospheric water budget. Q J R Meteorol Soc. 2018;144:1028–1044. https://doi.org/10.1002/qj.3273

Miglietta, M.M., Mastrangelo, D. and Conte, D. (2015) Influence of physics parameterization schemes on the simulation of a tropical-like cyclone in the Mediterranean sea. Atmospheric Research, 153, 360–375.

Miglietta MM, Carnevale D, Levizzani V, Rotunno R. Role of moist and dry air advection in the development of Mediterranean tropical-like cyclones (medicanes). Q J R Meteorol Soc. 2021; 147: 876–899. https://doi.org/10.1002/qj.3951

**MINOR POINTS**

1. **L47-48: Consider that the results in Zhang et al. (2020) are based on the relatively coarse ERA5 reanalysis, while the structure may change significantly for high-resolution runs. L48-50: Note that convection mainly occurs in the extra-tropical phase.**

   The paragraph has been updated, also incorporating suggestions from Reviewer #2:

*"While a medicane may follow an offshore trajectory and be relatively small in size, the geographically constrained nature of the Mediterranean basin still allows it to produce significant impacts (Scicchitano et al., 2021; Borzi et al., 2024). The spatial distribution of winds and precipitation, particularly in relation to complex terrain and landfall, has been the focus of extensive research due to its potential to intensify local hazards. Zhang et al. (2020) reported that rainfall totals increase from the centre to approximately 0.8° before decreasing; however, this pattern may be affected by the limited horizontal resolution (~30 km) of the ERA5 dataset used in their analysis. Recent findings by Dafis et al. (2020), which focus on convective activity within a 200 km radius of the cyclone centre, reveal that only a subset of Medicanes exhibit intense inner-core convection. Among these, persistent deep convection in the upshear quadrants emerges as a key driver of intensification."*

2. **L51: Lagouvardos not Lavaguardos.**

   Thank you.The reference to Lagouvardos has been corrected accordingly.

3. **L71: a short summary of the paper is missing here.**

   We fully agree that a concise summary of the manuscript structure is necessary for reader guidance.

4. **L113: what do you mean with "The selection of particle numbers ensured a balanced distribution across both grid points and vertical levels"?**

   FLEXPART conserves atmospheric mass by design, requiring sufficient particle representation across all vertical levels. To achieve this, 10 million air parcels were initially distributed homogeneously throughout the atmosphere, ensuring adequate sampling of both horizontal grid points and vertical layers (Fernández-Alvarez et al., 2023). The number of particles needed depends on the scientific objective: while a few million may suffice for global-scale transport statistics, higher resolution is essential for case studies of specific synoptic events (Pisso et al., 2019).

   We have revised the sentence accordingly to clarify this point:

   "Ensuring atmospheric mass conservation and full three-dimensional coverage of the model domain required selecting a number of air parcels that provided a physically consistent and representative distribution across all horizontal grid cells and vertical levels (Pisso et al., 2019; Fernández-Alvarez et al., 2023)."

   Fernández-Alvarez, J. C., M. Vázquez, A. Pérez-Alarcón, R. Nieto, and L. Gimeno, 2023: Comparison of Moisture Sources and Sinks Estimated with Different Versions of FLEXPART and FLEXPART-WRF Models Forced with ECMWF Reanalysis Data. J. Hydrometeor., 24, 221–239, https://doi.org/10.1175/JHM-D-22-0018.1.

   Pisso, I., Sollum, E., Grythe, H., Kristiansen, N. I., Cassiani, M., Eckhardt, S., Arnold, D., Morton, D., Thompson, R. L., Groot Zwaaftink, C. D., Evangeliou, N., Sodemann, H., Haimberger, L., Henne, S., Brunner, D., Burkhart, J. F., Fouilloux, A., Brioude, J., Philipp, A., Seibert, P., and Stohl, A.: The Lagrangian particle dispersion model FLEXPART version 10.4, Geosci. Model Dev., 12, 4955–4997, https://doi.org/10.5194/gmd-12-4955-2019, 2019.

5. **L130-131: what do you mean with *filter the centers* in "The cyclone phase space method (Evans and Hart, 2003) was employed to filter the tracked low-pressure centres"?**

We have revised the sentence:

"The cyclone phase space method (Evans and Hart, 2003) was employed to characterize and classify the structure of the tracked low-pressure centres."

6. **L210-211: the values do not correspond with those shown in Fig. 3b.**

Thank you for pointing this out. We will update Figure 3b to ensure that the values correspond accurately with the text.

7. **L217: 1008 hPa? Or 998 hPa?**

The in-text values are correct, and we will update the figure accordingly to reflect these accurate pressure values.

8. **L217: what does it mean "which further corroborates the observed profiles"?**

We have revised the sentence:

"On 19 September, at 0300 UTC, the pressure data from the Pylos buoy, although located farther from the cyclone center (76.7 km from the ERA5 track), still captured a decrease in MSLP, peaking at 1008 hPa. The MSLP profiles from the simulations show lower central pressures than those recorded by the Pylos buoy. Simulation-derived MSLP profiles exhibit lower central pressures than those measured by the buoy, reflecting the expected radial pressure gradient characteristic of intense cyclones."

9. **L251: the reasons for the development of the "low-PV values bubble" are not provided.**

Regarding the development of the "low-PV values bubble", we acknowledge that the underlying mechanisms were not explicitly detailed in the original text. Our simulations successfully reproduce this key feature of Ianos's evolution. A comprehensive analysis of this phenomenon is presented in Sanchez et al. (2024).

10. **L288: how are the pathways of precipitating particles selected in Fig. 6?**

We initially selected a subset of particles characterized by extended trajectory lengths. From this subset, we subsequently extracted a variable number of particles based solely on their visibility within the map, aiming to capture sufficient representation from different pathways while maintaining clarity for distinction.

11. **Figure 7 top panels: x and y are not shown in the Figure. If the x-axis is horizontal, I would expect it represents longitude not latitude, as in the paper.**

In Figure 7 (top panels), the x- and y-axes were inadvertently not labeled. The x-axis corresponds to longitude, while the y-axis corresponds to latitude, consistent with the rest of the paper. We will correct the figure labels and caption to accurately reflect this.

12. **Figure 8: it is not immediate to identify the time each panel refers to. Please specify it. Also, please add "H" and "L" to identify high and low pressure you are referring to in the manuscript.**

We will update Figure 8. The figure will also be simplified to accommodate the inclusion of new cases for improved clarity.

13. **L353: what do you mean with "daily MSLP"?**

The paragraph has been revised to clarify that moisture uptake is measured every 12 hours, not daily as originally proposed. Therefore, the MSLP corresponds to the instantaneous value at the start of each 12-hour moisture uptake period.

14. **L371, 372: southerly should be northerly.**

Revised.

15. **Figure 9: only the top panel is commented on.**

Thank you for your comment. The revised text now reads:

"Figure 9 presents the 6-hourly instantaneous moisture uptake along backward trajectories for Ianos on 17 September at 0600 UTC, corresponding to the timeframe shown in Figure 8d. In the initial 6-hour segment (Fig. 9a), moisture acquisition occurs predominantly over the Mediterranean Sea. Extending the analysis to 12-hour backward trajectories (Fig. 9b) reveals a pronounced moisture uptake over inland Libya, where the intensity of moisture gain increases. This inland region experienced a marked rise in low-level water vapor mixing ratio throughout the 18-hour period, closely aligned with prevailing surface wind patterns (Figs. 9d–f). The peak moisture uptake at 12 hours likely reflects a local accumulation of air parcel density (Fig. S12). Interestingly, moisture uptake diminishes at 18 hours (Fig. 9c), coinciding with a reduction in particle density within this moist environment (Fig. S12)."

Regarding the bottom panel of Figure 9, we acknowledge that its labeling was incorrect. This panel was actually intended to be removed and should not have been included, as our analysis relies exclusively on the low-level water vapor mixing ratio (middle panel) rather than the total Integrated Vapor Transport (bottom panel) to better represent the moisture conditions relevant to the backward trajectories.

---

## Author Comment (AC2)

**Response to Reviewer #2**

We sincerely thank the reviewer for their insightful and constructive feedback, which has greatly helped us improve the clarity and depth of our manuscript. Below, we address each point raised by the reviewer in detail. Reviewer comments are presented in **bold**, followed by our responses.

We would also like to respectfully highlight that Reviewer #1 recommended the inclusion of new results based on additional case studies and higher temporal resolution tests. While we fully acknowledge the importance and value of these suggestions, implementing them would require substantial computational resources and, more importantly, additional time beyond the current revision timeline. These extended simulations are currently being conducted as part of ongoing work, in parallel with awaiting the editorial decision on the present version.

We have made every effort to address all reviewer comments as thoroughly as possible within the scope of this revision. Concerning the specific suggestions related to the Results section, which would require significant additional analysis, we propose a clear plan to incorporate these new results in a revised version of the manuscript, should it be considered for further review. Some of the reviewer's detailed observations concerning the current version of the Results and Conclusion sections may no longer be applicable, since this parts of the manuscript will undergo substantial changes in a future revision. We remain fully committed to implementing these enhancements in a timely and rigorous manner.

**Major comments**

1. **The scope of the paper is blurred: what is known or not known about moisture sources in medicanes and how do they compare with Mediterranean cyclones in general? It is very surprising that a former similar study by the first author about moisture sources in a medicane is not mentioned (https://doi.org/10.3390/atmos13081327). Also, why focus on Ianos? For instance, the more recent Medicane Daniel also produced (even more?) extreme precipitation, over both the Balkans and North Africa, while the recent Valencia floods were unrelated to a medicane. The Introduction must be strengthened to clarify the scope, while the results must be brought in the context of the existing literature.**

   We sincerely thank the reviewer for their thoughtful and constructive feedback. We fully acknowledge the need to better delineate the scope of our study and more clearly position our findings within the broader context of existing literature on moisture sources in medicanes and Mediterranean cyclones. We have now revised the Introduction to explicitly frame our study within this context, as detailed in the revised lines referenced in the minor comments. We are also grateful for the reminder to reference our previous work.
   Regarding the focus on Medicane Ianos: while this event has indeed been the subject of multiple studies, we intended to revisit it using a state-of-the-art high-resolution modelling framework (WRF-FLEXPART) to reveal previously unresolved details, particularly with the vertical and temporal characteristics of moisture uptake. This case also benefits from the availability of high-quality observational data and satellite measurements, which make it particularly well-suited for analysis, especially regarding precipitation patterns and the

microphysical processes associated with convection. These motivations are now clearly articulated in the revised manuscript.

We agree with the reviewer that including more recent events, such as Medicane Daniel would help generalize the findings. In response to this and similar feedback from Reviewer #1, we have initiated high-resolution simulations for Medicanes Zorbas (2018) and Daniel (2023). These additional case studies will enhance the representativeness of our conclusions. However, due to the high computational demands of the moisture tracking simulations and the increased temporal resolution requested by Reviewer #1, even for a single phase of the cyclone's life cycle, the model must still be initialized ten days before the target event. As a result, we anticipate that completing the additional analyses will require more time. Pending editorial approval, we would be glad to incorporate these comparative results into the revised manuscript to further strengthen its scope and relevance.

2. **The methods are awkward in that they follow a systematic approach (automatic identification of all cyclones over a long time period and automatic adjustment of the cyclone extent) but are applied to a single case study. Furthermore, two sensitivity simulations are presented but not discussed despite surprising behaviour. Finally, the actual results related to moisture sources start in Section 4.2 only (wrongly labelled as 4.1; should actually be 3.2) and are not thoroughly developed and discussed. Altogether, this appears unbalanced and again questions the actual scope of the paper.**

We acknowledge that, in its present form, the Methods section may appear misaligned with the narrower scope of the analysis. However, the study is currently being expanded to include two additional medicanes, Zorbas (2018) and Daniel (2023), which are being simulated using the same high-resolution WRF-FLEXPART framework. This multi-case approach will align more closely with the systematic methodology described. Provided that the editor approves an extension to the revision timeline, the final version of the manuscript will reflect a more comprehensive framework, consistent with the study's design and capable of supporting more generalizable conclusions.

Regarding the two sensitivity simulations, we agree that they were insufficiently integrated into the overall discussion. These were originally intended as methodological checks; however, upon further inspection, we recognize that they exhibit notable and physically meaningful differences. In the revised manuscript, we will clearly state the rationale behind their design and include a focused discussion of the key results, emphasizing their relevance to the interpretation of the primary case and the robustness of our conclusions.

We also thank the reviewer for pointing out the mislabeling of the moisture source results section. As noted, what was labeled Section 4.2 should have been Section 3.2, and this has been corrected in the revised version.

To address the concern regarding imbalance and underdevelopment of key results, we are restructuring the Results section to better emphasize the moisture source analysis. The forthcoming inclusion of Zorbas and Daniel will allow for a broader yet targeted comparative assessment of moisture uptake across different medicanes. This expansion will not only provide greater depth but also ensure that the analysis better reflects the systematic nature of the methodology.

**Minor comments**

1. **24 Fig 1b**

   The suggested correction has been implemented in the revised manuscript.

2. **34-37 It is unclear how this result compares with the above l. 33-34**

   We have revised lines 30–37:

   *"Moisture sources for cyclone precipitation have been extensively investigated, revealing significant contributions from the Mediterranean Sea alongside remote sources such as the tropical and extratropical Atlantic Oceans and tropical Africa, with considerable variability in moisture origin patterns (Winschall et al., 2014; Chazette et al., 2016; Lee et al., 2017; Raveh-Rubin and Wernli, 2016; Duffourg et al., 2018). Through detailed Lagrangian analysis of five intense Mediterranean cyclones, Raveh-Rubin and Wernli (2016) showed that these systems draw moisture both locally from the Mediterranean and from diverse remote regions, with notable inter-event variability. This inter-event variability in moisture sourcing was further confirmed by Winschall et al. (2014) through analysis of a larger set of cyclones under similar large-scale conditions. Building on these findings, Flaounas et al. (2019) expanded the analysis to an even broader dataset, demonstrating that cyclones producing heavy rainfall tend to receive increased moisture influx particularly from the eastern Atlantic and western Mediterranean basin, thereby highlighting the critical role of moisture source diversity in controlling precipitation intensity."*

3. **45 are there driving processes other than diabatic forcing and baroclinic instability?**

   In the introduction, we highlighted the main drivers, but the complexity is increasing since this is mainly studied on a case-by-case basis. We have added the following clarification:

   *"In addition, mesoscale processes such as air–sea interactions, deep convection, and orographic forcing can exert a critical influence on the development and intensification of certain medicanes, particularly during their mature stages as discussed in detail by Flaounas et al. (2021, Section 4.2). Prominent examples include Medicane Rolf (6–9 November 2011; Miglietta et al., 2013; Dafis et al., 2018, 2020), the cyclone of 24–26 September 2006 (Moscatello et al., 2008), the October 1996 event (Mazza et al., 2017), and Medicane Qendresa (7–8 November 2014; Bouin and Lebeaupin-Brossier, 2020)."*

4. **47 "They" = medicanes? Flaounas et al. 2018 refer to Mediterranean cyclones in general**

   We have clarified this point in the updated version of the manuscript:

   *"Although Medicanes rarely exceed the intensity of a Category 1 hurricane on the Saffir–Simpson scale (Miglietta and Rotunno, 2019), they are nonetheless associated with strong winds, intense rainfall, high wave activity, and storm surges, all of which can pose significant hazards to densely populated coastal regions of the Mediterranean (Carriò et al., 2017; Dafis et al., 2018; Di Muzio et al., 2019; Bouin and Brossier, 2020; Portmann et al., 2020; Faranda et al., 2022; Lagouvardos et al., 2022; Varlas et al., 2023)."*

5. **47–52 values would be helpful here: how close is close, near, or proximity? Also, ERA5 used in Zhang et al. 2020 is limited by its horizontal resolution (about 30 km)**

Lines have been updated:

"*While a medicane may follow an offshore trajectory and be relatively small in size, the geographically constrained nature of the Mediterranean basin still allows it to produce significant impacts (Scicchitano et al., 2021; Borzi et al., 2024). The spatial distribution of winds and precipitation, particularly in relation to complex terrain and landfall, has been the focus of extensive research due to its potential to intensify local hazards. Zhang et al. (2020) reported that rainfall totals increase from the centre to approximately 0.8° before decreasing; however, this pattern may be affected by the limited horizontal resolution (~30 km) of the ERA5 dataset used in their analysis. Recent findings by Dafis et al. (2020), which focus on convective activity within a 200 km radius of the cyclone centre, reveal that only a subset of Medicanes exhibit intense inner-core convection. Among these, persistent deep convection in the upshear quadrants emerges as a key driver of intensification.*"

6. **51 Lagouvardos**

Thank you.The reference to Lagouvardos has been corrected accordingly.

7. **52–54 this last sentence appears disconnected from the paragraph**

Thank you for the clarification. Here's a polished version of the revised paragraph, written as a new standalone paragraph following the response to Comment #5:

"*Conversely, Dafis et al. (2020) identified a subset of medicanes that underwent significant intensification despite the presence of only weak or sporadic deep convection near their centers. In these cases, the authors hypothesized that deep convection may play a secondary role in the intensification process. Lagouvardos et al. (2021) analyzed the evolution of the intense Medicane Ianos (September, 2020) and similarly reported a temporary weakening of convective activity prior to its transition into a tropical-like phase. Nevertheless, it achieved a structural organization comparable to other medicanes whose intensification was accompanied by persistent deep convection, particularly in the upshear quadrant (e.g., Trixie and Zorbas; Dafis et al., 2020; Lagouvardos et al., 2021).*"

8. **56 reference?**

We have added the appropriate reference: Lagouvardos et al. (2021).

9. **69–71 more motivation for the study is needed here: in which sense was the precipitation associated with Ianos unprecedented, and why does it matter?**

"*The evolution of Medicane Ianos presents a distinctive case within the spectrum of Mediterranean tropical-like cyclones. Its development was characterized by initially weak deep convection prior to intensification, similar to Medicane Qendresa (Dafis et al., 2020), and by a later phase marked by high symmetry and intensity, as observed in Medicane Zorbas (Lagouvardos et al., 2021). Notably, Ianos exhibited cloud-top heights surpassing those recorded for Medicane Numa (Marra et al., 2019). This rare evolution was fortuitously captured by satellite overpasses and has been extensively analyzed (D'Adderio et al., 2022). However, the associated water budget, particularly relevant for understanding precipitation dynamics (Lagouvardos et al., 2021), remains poorly constrained. Our*"

*objective is to identify the Lagrangian moisture sources that contributed to Ianos's development and to determine whether these pathways are shared with or distinct from those of other medicanes. To this end, we employ high-resolution simulations that have undergone prior sensitivity testing, with the goal of advancing both our understanding of moisture transport dynamics and the modeling fidelity of such events in the Mediterranean region.*"

The highlighted objective is subject to the editor's discretion.

**10. 71 a short summary is typically expected here (Section 2 shows this, Section 3 shows that, …)**

We fully agree that a concise summary of the manuscript structure is necessary for reader guidance.

**11. 80 strictly speaking it is a 24-day period**

This has been corrected accordingly.

**12. 80 why the model spin-up? (without requiring the reader to dig into the references)**

Regarding the model spin-up, the text has been revised for clarity as follows:

"To allow the model to internally adjust from the initial atmospheric and surface conditions toward a dynamically balanced and physically consistent state."

**13. 83 the configuration sounds unusual: 18 km is quite close to the resolution of ERA5, while 6 km lies in the grey zone of deep convection; some discussion is needed here**

We agree that this setup, where the outer domain (18 km) is relatively close to the ERA5 native resolution and the inner domain (6 km) falls within the grey zone of deep convection, merits further clarification.

This configuration was selected as a compromise between the need to resolve key mesoscale features and the practical constraints of computational cost. The 6 km resolution enables improved representation of topography-driven circulations, mesoscale convergence zones, and organized convection, which are essential to medicane development. Although convection at this scale is only partially resolved, fully convection-permitting resolutions (<3 km) would drastically increase computational demands, particularly given that Lagrangian moisture source attribution with FLEXPART-WRF requires running high-resolution, multi-day particle tracking simulations.

To address the limitations associated with the grey zone, we employed the Kain–Fritsch convective parameterization scheme in WRF, a widely used approach for medicanes that has been validated in prior literature (e.g., Miglietta et al., 2015; Fita and Flaounas, 2018; Miglietta et al., 2021). In parallel, FLEXPART was configured to activate both turbulence and convection parameterizations, following recommendations in Brioude et al. (2013), to ensure physical consistency between advective and subgrid transport processes. We enabled the Hanna turbulence scheme (Hanna, 1982), which calculates turbulent mixing in the planetary boundary layer using WRF-derived fields such as the PBL height, Monin–Obukhov length, friction velocity, and convective velocity scale.

**14. l01 missing reference Beck et al; what is the resolution of ERA5 and MSWEP?**

We have included the citation for Beck et al., as well as the resolutions of ERA5 (0.25°) and MSWEP (0.1°), in the revised manuscript.

**15. 111 Fig 1a; is the yellow box different from the blue box?**

Yes, the yellow box in Fig. 1a represents the domain used for the FLEXPART-WRF simulations, while the blue box corresponds to the larger parent WRF domain.

**16. 115–120 without prior knowledge of FLEXPART it is not fully clear why the mentioned variables and processes are not taken from WRF**

We agree that the explanation regarding the necessity of activating turbulence and convection parameterizations in FLEXPART could be clearer for readers unfamiliar with the model. Similar to our response to comment #13, we have expanded the manuscript to clarify this point. Specifically, we have added the following text:

*"Following recommendations from the FLEXPART-WRF framework (Brioude et al., 2013), without turbulence parameterization, FLEXPART operates as a non-dispersive Lagrangian trajectory model. Using the Hanna turbulence scheme (Hanna, 1982), FLEXPART internally computes planetary boundary layer (PBL) turbulent mixing based on WRF-derived parameters such as PBL height, Monin–Obukhov length, convective velocity scale, roughness length, and friction velocity."*

**17. 122–133 the approach is certainly relevant for a systematic identification but as a single cyclone is investigated here the details of automatic filtering are not needed and the CyTRACK reference is sufficient**

The methodology described forms the foundation for our ongoing expansion toward a multi-case study framework.

**18. 133 ..**

**19. 140–158 repetitions between the text and the (long) figure caption**

In the revised manuscript, we have streamlined the content to avoid unnecessary repetition. We have shortened the figure caption to focus on essential visual elements

**20. 175 Stohl and James**

Resolved.

**21. 177–179 similar to l. 115–120: why not use precipitation and cloud microphysics information from WRF?**

We appreciate the reviewer's question. Unlike Eulerian methods that rely on grid-based precipitation or microphysical fields directly from WRF, our Lagrangian approach using FLEXPART-WRF tracks individual parcels of air. Moisture source attribution is based on changes in specific humidity along each particle's trajectory. By focusing on particles that ultimately contribute to precipitation, our study specifically examines moisture associated with rainfall. Accordingly, the identification of precipitating particles or air masses is based on the loss of specific humidity along their Lagrangian trajectories.

**22. 183–186 is FLEXPART-WRF or TROVA used here? Or both?**

Typically, software tools are used to execute the Lagrangian tracking methodology. Once the outputs from a dispersion model (e.g., FLEXPART, FLEXPART-WRF, or LAGRANTO) are available, these tools facilitate the subsequent steps: defining the target region, selecting precipitating particles, and backtracking them to identify moisture sources. Software such as TROVA enables users to specify parameters such as the target region, tracking duration, and thresholds for identifying precipitating particles through a user-defined input file.

**23. 187 Where is Section 3?**

This was a typographical error.

**24. 189–193 the purpose of the sensitivity tests is not fully clear; for instance, it would be helpful to know more about the outcome of these papers and how they compare with yours**

We appreciate this observation. In the revised manuscript, we have clarified the purpose and relevance of the sensitivity tests. The improved Introduction now more clearly outlines the scope of our study and the role of these tests in validating and contextualizing our results.

**25. 194–197 this information would be useful in Section 2.1**

We agree with the reviewer, and the information has now been moved to Section 2.1

**26. 198–205 please specify which curve to look at (the simulation names are not very human-readable)**

In the revised manuscript, we will clarified this aspect.

**27. 207–208 in which sense is it in closest agreement? As shown below, ERA5 completely underestimates the observed intensity, which is actually well captured by WRF**

Thank you for pointing this out. We clarify in the revised manuscript that the "closest agreement" refers to the relative performance among different WRF configurations.

**28. 209–211 as stated in the introduction**

True.

**29. 212 which time lag?**

The time lag refers to the difference between the minimum mean sea level pressure (MSLP) of Ianos in the 6-hourly WRF simulations and the recorded minimum MSLP at surface weather stations, which have hourly resolution. This sentence has been revised in the manuscript to clarify this point.

**30. 213 the camel-shaped time evolution in WRF_full_ndg and WRF_deact_ndg deserves some comments (or should be removed)**

Actually, these results reflect the trajectories obtained from the outputs of these configurations, where at the earlier stages Ianos is positioned away from its best track. This will be discussed in the manuscript.

31. **215–217 unclear**
We will revise these sentences to clarify the intended meaning.

32. **219 simulation names in the supplementary are not consistent with the main paper and only two are shown**
We acknowledge the inconsistency and will correct the simulation naming across the supplementary material. Additional simulation outputs will be included in the Supplement..

33. **220 "most accurate" compared to ERA5? And in which sense?**
We will specify the metrics used to assess accuracy (e.g., track position, intensity, wind fields) and explicitly state that the comparison is relative to ERA5, where applicable.

34. **221 the deep warm core structure of WRF_FUL deserves some comments (or should be removed)**
We agree this deserves further commentary. We will either elaborate on this feature.

35. **237 it is not clear what should be learned from Section B in the Supplementary Material, and only one simulation is shown (which name is not consistent with the main paper)**
We acknowledge that the objective of Section B in the Supplementary Material is not sufficiently clear. This section primarily supports the decision to select the configuration with nudging above the planetary boundary layer (PBL). We will revise the manuscript to better reference this section and clarify its role in justifying our model setup.

36. **238–239 more details are needed to support the claim that the wind field is accurate (when and where in the simulation and compared to which observations); moreover, the 6 h intervals are clearly not sufficient to depict the simulated wind footprint, which shows 'jumps' in Fig. 3**
We will improve the clarity of this part by either moving it to an appendix or developing it into a dedicated subsection. The meteorological variables, including wind, were statistically and significantly validated over the entire 24-day simulation period, as detailed in Section B of the Supplementary Material. The apparent 'jumps' in the wind footprint shown in Fig. 3 result from how the figure was composed, depicting the wind within the cyclone's radius at each cyclone position. We will revise the figure caption and corresponding text to better explain this visualization choice.

37. **240 which time step?**
We will specify the exact time step being referred to.

38. **243 a number would be helpful**
We will include a specific numerical value 1000 to 1400 kg m-1 s-1.

39. **245–274 the dynamics of cyclone Ianos are largely described in the aforementioned references; it is not clear how their results compare with yours and what is new here**
Our simulations, once again, offer strong evidence of well-resolved dynamics of Ianos.

40. **256 missing reference Flaounas et al. 2021; and "major medicane" is not common terminology**
We have included the missing reference to Flaounas et al. (2021) and rephrased "major medicane" to use standard terminology.

41. **259–260 in the selected WRF simulation?**
We will clarify that this refers to the selected WRF simulation and explicitly name it in the text, although this was already mentioned in lines 233–235.

42. **266 maximum vertical extent?**

Revised.

43. **267 is convection shown somewhere?**

Although our study does not explicitly show convection, Fig. 7 is related to this behavior. These results are consistent with the convective patterns reported by Lagouvardos et al. (2021), and we will clarify this connection in the revised manuscript.

44. **269–271 in the selected WRF simulation?**

We will clarify that this refers to the selected WRF simulation.

45. **275 Section 4.2**

Revised.

46. **276–281 the description suggests independent sources of moisture, which does not match the continuous area suggested by Fig. 6a; this mismatch may be due to the map rendering that hardly allows identifying coasts**

In Fig. 6a, we present the total moisture uptake associated with Ianos over its entire lifetime. Our aim is not to suggest isolated or independent sources of moisture. Referring to these regions individually provides a more accurate and nuanced interpretation of the spatial distribution, which would be obscured by a single, generalized label. Moreover, Figs. 6b–e display trajectories of selected air masses at specific stages of the cyclone's evolution, further supporting the identification and interpretation of moisture uptake regions.

47. **288–297 geographical labels (as in Fig. 1b) or numbering i) ii) iii) on Fig. 6 would be helpful to follow the discussion**

We will improve the figure.

48. **312–313 is this criterion different from the precipitation trajectories in Fig. 6?**

The criterion mentioned here is not different from the precipitation trajectories shown in Fig. 6.

49. **319–320 more details are needed on these satellite measurements of Ianos (what, where and when?)**
**323–324 what type of satellite data?**

We thank the reviewer for these observations. In the revised manuscript, we have clarified that the satellite data mentioned (from SEVIRI, scatterometers, and the GPM Core Observatory, including the DPR and GMI instruments) were not directly analyzed in our study. Instead, we refer to these datasets in the context of findings from previous studies specifically dedicated to the satellite-based observation of Medicane Ianos (Lagouvardos et al., 2021; D'Adderio et al., 2022). We have now added information about the temporal context of these satellite measurements as documented in those works.

50. **327 where and when? Missing reference Hourngir et al., 2021**

We have include the missing reference and:

"In the development phase of Ianos on 16 September at 13:40 UTC (D'Adderio et al., 2022)."

51. **331 what is new compared to Fig. 6?**

Fig. 6a presents the total moisture uptake integrated over the entire lifetime of the cyclone,

providing a comprehensive overview of moisture sources throughout the cyclone's evolution. Figs. 6b–e present the trajectories of air parcels located within the cyclone radius that are directly associated with precipitation during the intensification and mature stages. In contrast, Fig. 7 refines this analysis by illustrating the moisture uptake field specifically for the intensification and mature phases. This distinction will be more carefully handled and clearly explained in the revised version of the manuscript.

52. **332 "Uptake" is a noun**
We revised the sentence to ensure proper grammatical usage.

53. **336–338 this information would be helpful in the introduction**
We agree and will move this contextual information to the Introduction.

54. **339 focussing the map on the Mediterranean would be more relevant (and readable)**
We will adjust the map extent to better center on the Mediterranean and enhance readability.

55. **Fig. 8 is quite busy and the MSLP and VIMF fields are quite noisy, which make the discussion hard to follow**
We will simplify or reformat Fig. 8.

56. **351–352 this information would be helpful in the introduction**
We will consider relocating this background to the Introduction for better context-setting.

57. **354 acronym VIMF is not needed as not used**
We removed the acronym.

58. **366-367 where is the chanelling or low-level jet to be seen in Fig. 8?**
We refer to the wind direction and spatial pattern rather than the presence of a distinct low-level jet. If the term "channeled" caused any confusion suggesting a low-level jet, we are happy to revise the wording for clarity.

59. **375–379 it is unclear what to learn from Fig. 9, as only panel (a) is referred to**
We will revise the text to refer to all relevant panels in Fig. 9 and clarify the purpose of this figure.

60. **393–394, 426 "high resolution" is disputable for convective precipitation, which simulation typically requires horizontal grid spacing of O(1 km).**
We respect and appreciate this point of view.

61. **396, 398, 402 it is not common practice to refer to specific figures in the conclusions, especially to the supplementary material.**
We will remove or rephrase these references to align with standard practice in conclusions.

62. **what is an "improved" cyclonic organization?**
It was revised to "A well-defined cyclonic circulation".